# From Diffusion to Rectified Flow: Rethinking Text-Based Segmentation

## Abstract

Text-based image segmentation aims to delineate object boundaries within an image from text prompts, offering higher flexibility and broader application scope compared to traditional fixed-category segmentation tasks. Recent studies have shown that diffusion models (e.g., Stable Diffusion) can provide rich multimodal semantic features, leading to studies of using diffusion models as feature extractors for segmentation tasks. Such methods, however, inherit the generative natures of diffusion models that are harmful to discriminative segmentation tasks. In response, we propose RLFSeg, a novel framework that leverages Rectified Flow to learn direct mapping from the image to the segmentation mask within the latent space. The model is thus freed from the noise-denoise process and the need to optimize the time step of diffusion models, resulting in substantially better performance than previous diffusion-based methods, especially on zero-shot scenarios. By introducing label refinement and an Adaptive One-Step Sampling strategy, the model achieves higher accuracy even on a single inference step. The framework redirects a pretrained generative model to the discriminative segmentation task with zero modification to model structure, thus reveals promising application potential and significant research value.

## 1 INTRODUCTION

Text-based image segmentation aims to accurately delineate object boundaries in an image according to a given textual prompt. Existing unsupervised or weakly supervised methods often rely on network-level annotated datasets and specially designed models to achieve precise segmentation. With the development of Latent Diffusion Models (LDMs) Pnvr et al. (2023), impressive results have been demonstrated in text-to-image generation. Prior studies Pnvr et al. (2023) have shown that LDMs inherently encode rich instance-level text–image alignment, which has sparked growing interest in extending their use beyond generation tasks toward semantic segmentation.

For segmentation tasks based on diffusion models, the prevailing approach Pnvr et al. (2023) involves freezing the diffusion-related modules to serve as a feature extractor, with a final head layer producing the segmentation output. While this strategy enables the model to quickly acquire multimodal knowledge and demonstrates strong generalization, it often yields coarse masks with imprecise boundaries. This limitation arises from a fundamental mismatch between the generative nature of diffusion processes and the discriminative nature of image segmentation. Generative modeling emphasizes diversity, where a single input may correspond to multiple acceptable outputs. In contrast, segmentation requires determinism, embodying a one-to-one mapping where a specific image and query must correspond to a single, well-defined mask.

Rectified Flow (RF) addresses this challenge by learning a deterministic, near-linear Ordinary Differential Equation (ODE) trajectory between the source and target domains. This property aligns naturally with the requirements of image segmentation, making RF particularly well-suited for cross-task adaptation. As a result, it establishes a solid theoretical foundation for using diffusion models as backbones, enabling a smoother and more principled transition from generative to discriminative tasks.

Based on the observations above, we propose a novel framework, named RLFSeg, that leverages the strengths of Latent Diffusion Models (LDMs) to generate high-quality and precise segmentation masks. Our framework consists of three key components: Rectified Latent Flow, Refinement

(a) LD-ZNet framework        (b) Our framework

Figure 1: Prior methods rely on LDM as a feature extractor with extra branches, while ours directly enables text-based segmentation via finetuning.

and Dynamic Selection, and Adaptive One-Step Sampling. Existing methods often rely on stepwise denoising or additional UNet branches to extract latent features for mask generation, which incurs extra computational cost and risks error propagation. As shown in Figure 1, unlike previous approaches that use the model as a backbone, Rectified Latent Flow directly transforms image latents to mask latents in a single step, efficiently capturing semantic guidance from textual prompts. To further enhance mask quality and reduce the impact of annotation noise, we introduce a refinement and dynamic selection module that iteratively sharpens object boundaries and adaptively alternates between original and refined labels during training. Finally, our adaptive one-step sampling mechanism dynamically scales the latent update to ensure accurate boundary coverage within a single-step sampling process. By integrating these components, our framework produces segmentation masks that are both semantically aligned with the text input and visually precise, while remaining computationally efficient and robust to noisy annotations.

Extensive experiments demonstrate the effectiveness of our proposed framework. Our method achieves state-of-the-art results on multiple text-to-image segmentation benchmarks, including PhraseCut Wu et al. (2020), RefCOCO Kazemzadeh et al. (2014), RefCOCO+ Kazemzadeh et al. (2014), and G-Ref Nagaraja et al. (2016). In summary, the main contributions of this work are as follows:

- We introduce Rectified Latent Flow to reconcile the generative nature of diffusion with the deterministic demands of segmentation by learning a direct, latent image-to-mask transformation.
- We introduce label Refinment and Dynamic Selection (RDS) module to iteratively improve mask quality and mitigate annotation noise.
- We design an Adaptive One-step Sampling (AOS) mechanism, improving boundary accuracy and overall mask precision in a single step.
- Extensive experiments demonstrate that our method achieves state-of-the-art performance in both mIoU and AP.

In the spirit of transparency, we state that the Gemini model was utilized to refine the phrasing of this paper for enhanced readability.

## 2 RELATED WORK

### 2.1 TEXT-BASED IMAGE SEGMENTATION

Text-based image segmentation aims to create pixel-level masks from free-form text, offering flexibility beyond fixed categories by handling both "stuff" and "instances". The field has evolved through several paradigms. Early works fused features from RNN and CNN backbones Hu et al. (2016); Li et al. (2018); Shi et al. (2018); Ye et al. (2019), later enhanced by attention mechanisms for better cross-modal alignment Margffoy-Tuay et al. (2018); Wang et al. (2022); Yu et al. (2018). A significant shift occurred with large-scale models like CLIP Radford et al. (2021), which improved representation learning and led to powerful foundation models such as SAM Kirillov et al. (2023) and SEEM Zou et al. (2023) with strong zero-shot capabilities. More recently, the trend has moved towards integrating segmentation into Vision-Language Large Models (VLLMs) for conversational

reasoning. These models evolved from coarse bounding box grounding Chen et al. (2023); You et al. (2023) to direct mask prediction Lai et al. (2024); Ren et al. (2024); Rasheed et al. (2024). However, despite their progress, these discriminative approaches often exhibit significant limitations. Many struggle to generate highly precise boundaries for complex, free-form instructions, while others require complex architectural modifications and costly fine-tuning to adapt to new tasks. This motivates exploring alternative generative paradigms, which may offer a more principled and effective approach to this task.

### 2.2 TEXT-TO-IMAGE SYNTHESIS

Text-to-image (T2I) synthesis has advanced rapidly from early GAN-based Xu et al. (2018); Zhu et al. (2019); Tao et al. (2022); Zhang et al. (2021); Ye et al. (2021); Zhou et al. (2022) and autoregressive Ramesh et al. (2021); Ding et al. (2021); Gafni et al. (2022) models with vector-quantized autoencoders Van Den Oord et al. (2017); Razavi et al. (2019); Esser et al. (2021) to diffusion models Nichol & Dhariwal (2021); Dhariwal & Nichol (2021), which significantly improve image quality and diversity. Pixel-space diffusion models are computationally expensive, motivating latent diffusion models (LDMs) Nichol et al. (2021); Gu et al. (2022); Tang et al. (2022); Rombach et al. (2022) that enable efficient high-resolution synthesis. Large-scale models such as Stable Diffusion Esser et al. (2024); Podell et al. (2023) and Imagen Baldridge et al. (2024), along with recent transformer-based architectures Black Forest Labs (2024); Podell et al. (2023); Peebles & Xie (2023), further enhance photorealism and scalability. These developments establish diffusion models as a dominant T2I paradigm and provide rich semantic features useful for downstream tasks such as text-based image segmentation.

### 2.3 GENERATIVE MODELS FOR TEXT-BASED SEGMENTATION

The remarkable scalability and transferability of diffusion models Nichol & Dhariwal (2021) make them a promising foundation for segmentation. Early work showed that features from generative models could be repurposed for this task Baranchuk et al. (2021), though often in limited few-shot Fei-Fei et al. (2006) or domain-specific settings Karras et al. (2019); Yu et al. (2015). More recently, diffusion models have been adapted for text-driven segmentation via two main strategies. Training-free methods Corradini et al. (2024); Karazija et al. (2023) align internal features with text but yield coarse boundaries, as the features are optimized for generation. To improve precision, training-based adaptations are used, but they often introduce significant overhead through complex multi-stage pipelines Li et al. (2023), auxiliary modules, or costly alignment training Pnvr et al. (2023); Stracke et al. (2025). These strategies treat the diffusion model as a component rather than reframing its core process for segmentation. In contrast, our method offers a more fundamental solution by employing Rectified Flow to reframe the task. We directly fine-tune a pretrained LDM to learn a deterministic, single-step mapping from image to mask, effectively transforming the stochastic, multi-step generation process and achieving superior segmentation performance.

## 3 METHOD

In this section, we first introduce the preliminary knowledge required for understanding the key components of our method. We then detail our proposed framework, RLFSeg, which enhances text-to-image segmentation by leveraging the strengths of Latent Diffusion Models (LDMs). Our method consists of three core components: 1) Rectified Latent Flow, which refines latent flow to directly generate segmentation masks from the original image in a single step; 2) Refinement and Dynamic Selection(RDS), which utilizes the Segment Anything Model (SAM) Kirillov et al. (2023) for label optimization and automatic loss selection; 3) Adaptive One-Step Sampling(AOS), which dynamically adjusts the norm of the predicted velocity $v$, enabling our approach to achieve high-quality results in a single sampling step. The pipeline of our method is illustrated in Figure 2.

### 3.1 PRELIMINARIES

**Latent Diffusion Models**(LDMs) Rombach et al. (2022) generate images in a compressed latent space through two main stages. First, an autoencoder (e.g., VQGAN Esser et al. (2021)) maps an input image $x$ to a latent representation $z = \Phi_{encoder}(x)$, preserving its semantic content in a

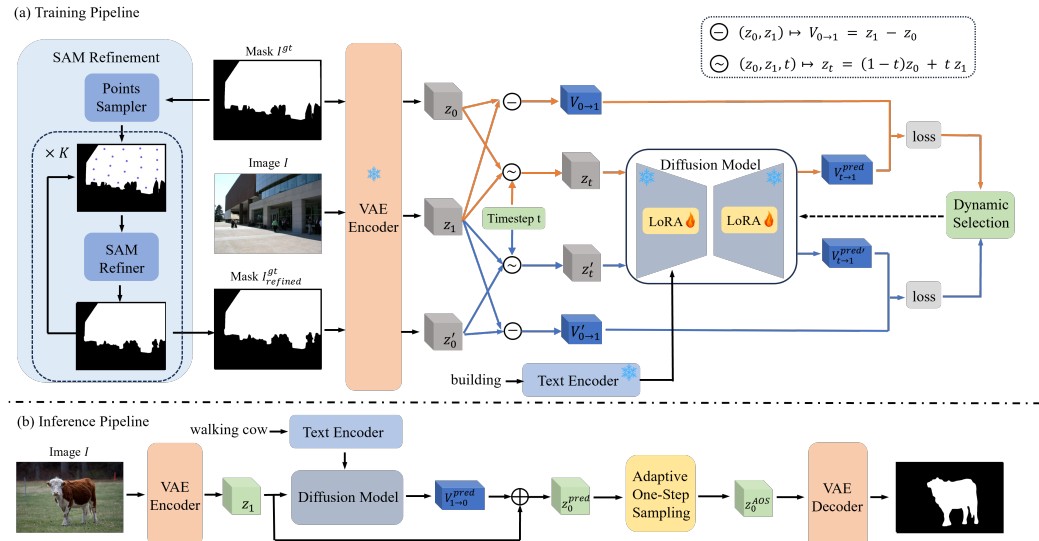

Figure 2: **Overview of RLFSeg.** (a) Training pipeline with Rectified Latent Flow and SAM-driven Label Refinement, where ground-truth labels are automatically matched with SAM-refined annotations for finer supervision by iteratively sampling points to progressively refine masks. (b) Inference pipeline with Adaptive One-Step Sampling to dynamically adjust the step size for sampling.

compact form. Then, a diffusion UNet learns to iteratively denoise the latent via a reverse process, guided by text features extracted from a pretrained CLIP encoder Radford et al. (2021) through cross-attention. The denoising process can be written as

$$z_t = f_\theta(z_{t-1}, t, c), \tag{1}$$

where $z_t$ is the latent at step $t$, $f_\theta$ is the denoising function, and $c$ denotes the text condition. This formulation enables efficient and high-quality text-to-image generation.

## 3.2 RECTIFIED LATENT FLOW

The training paradigm of diffusion models, such as DDPM, is centered around a progressive noising process on target data to construct a generative path from pure noise to real data. This stochastic perturbation mechanism is the fundamental reason for the diversity in the outputs of diffusion models. However, an inherent task conflict arises when this paradigm is directly applied to image segmentation. The conventional practice involves conditioning on the source image while modeling the noised mask, which essentially forces a discriminative task that seeks a unique, deterministic solution into a generative framework designed for diversity. For segmentation, a given image and text prompt should map to a single, deterministic mask. Therefore, noising the mask to imitate the generation process is a circuitous and unnecessary redundant design.

The emergence of Rectified Flow provides an elegant solution to this cross-task alignment dilemma. As a new-generation generative paradigm, it aims to learn a deterministic, near-linear Ordinary Differential Equation (ODE) path between source and target domains, which is theoretically consistent with the optimal transport (OT) path connecting the image and the mask. This mechanism obviates the need for stochastic noising. Given the inherent compatibility between the determinism of Rectified Flow and the intrinsic demands of segmentation, we recognized that it provides a solid theoretical bridge for leveraging the powerful pre-trained knowledge of diffusion models. To this end, we propose RLFSeg, a method designed to directly learn a continuous, direct mapping path from the source image to the target mask, thereby seamlessly aligning the powerful generalization capabilities of the generative model with the objectives of the discriminative task.

Given an input image $I$ and a mask $M$, we first use the VAE encoder $\Phi_{encoder}$ to obtain their latent representations, $z_1 = \Phi_{encoder}(I)$ and $z_0 = \Phi_{encoder}(M)$. We then extract frozen CLIP text features from the provided text prompt, which are fed into the denoising UNet of the LDM. This

setup allows the model to generate segmentation masks by leveraging semantic guidance from the text input.

During training, we directly learn the vector field $\mathbf{v} = z_1 - z_0$ that defines the straight path between the image latent $z_0$ and the mask latent $z_1$. Following Rectified Flow, we sample a time $t$ from a uniform distribution $U(0, 1)$ and construct an intermediate latent $z_t = tz_1 + (1 - t)z_0$. In summary, the Rectified Flow loss is defined in Eq.2:

$$\mathcal{L}_{rf} = \min_v \int_0^1 E\left[\|(z_1 - z_0) - v_\theta(z_t, t)\|^2\right] dt, \tag{2}$$

where $v_\theta$ is the model which is trained to predict the constant vector field $\mathbf{v}$ given the interpolated latent $z_t$ and timestep $t$.

### 3.3 REFINEMENT WITH DYNAMIC SELECTION

Segmentation masks in our dataset are generated from polygon-based annotations, which are often coarse and prone to noise, limiting their effectiveness for precise text-based segmentation. This issue is particularly exacerbated by the training methodology of Rectified Flow, as it makes the model more susceptible to learning the annotation style of the dataset. We provide several visual examples of this phenomenon in Appendix A.1. To address this, we propose a SAM-driven Label Refinement and Dynamic Selection Module that iteratively improves mask quality and adaptively leverages both original and refined labels during training.

**SAM-driven Label Refinement** iteratively applies SAM to refine the boundaries of segmentation masks. To provide the SAM model with guiding prompt, we generated a set of N points $P$ by applying k-means clustering to the initial mask via Eq. 3 . Notably, these points remain fixed throughout the iterative process, thus serving as spatial anchors that effectively reduce the cumulative positional drift of the mask. The algorithm is demonstrated in Algorithm 1.

$$\mathbf{P} = \text{KMeans}\left(\{(x, y) \mid \mathbf{M}_0(x, y) > 0.5\}, N\right), \tag{3}$$

where $\mathbf{M}_0(x, y)$ represents the value of the pixel at coordinates $(x, y)$ in the ground-truth mask.

In each iteration, SAM (denoted by $\Phi_{SAM}$) predicts a new mask $M_t$ based on the previous mask $M_{t-1}$ and the anchor points $P$. The process terminates once the IoU score converges. Through this process, the mask boundary is progressively refined to more accurately match the object's contour.

---

**Algorithm 1** Iterative Mask Refinement with Early Stopping

**Require:** $M_0$, $P$, $T_{max}$, $\tau$
**Ensure:** $M_{\text{refine}}$
 1: **for** $t = 1 \rightarrow T_{max}$ **until** $\text{IoU}(M_t, M_{t-1}) < \tau$ **do**
 2:      $M_t \leftarrow \Phi_{\text{SAM}}(P, M_{t-1})$
 3: **end for**
 4: $M_{\text{refine}} \leftarrow M_t$

---

**Dynamic Selection**, as defined in Eq. 4, automatically selects between the losses computed on the original mask and the refined mask. This strikes a crucial balance between the stability of the original annotations and the precision of the refined results. As our ablation studies in Section 4.3 will demonstrate, this approach yields significant performance gains.

$$\mathcal{L}_{final} = \mathbf{1}_{rds}\mathcal{L}_{rf}(z_0 = \mathbf{M}_0) + (1 - \mathbf{1}_{rds})\mathcal{L}_{rf}(z_0 = \mathbf{M}_{refine}), \tag{4}$$

where $\mathbf{1}$ is an indicator function that equals 1 if $\mathcal{L}_{rf}(z_0 = \mathbf{M}_0) < \mathcal{L}_{rf}(z_0 = \mathbf{M}_{refine})$, and 0 otherwise. The design follows a core logic: a higher-quality annotation simplifies the learning task, providing a clearer path for the model to converge to a lower loss. Although this scenario is relatively uncommon (occurring in approximately 15% of our training cases), when the refined annotation $M_{refine}$ paradoxically degrades in quality, the model will instead learn from the original label $M_0$.

Figure 3: **Comparison of sampling trajectories and the effect of AOS.** Interpolation with (**a**) the ground-truth $z_0^{truth}$ versus (**b**) our prediction $z_0^{pred}$. The artifact-laden $z_0^{pred}$ misaligns with the final target $z_0^{truth}$, corresponding instead to an intermediate ground truth $z_t^{truth}$ (e.g., $t = 0.1$). Our AOS corrects this misalignment by adaptively scaling the update step, yielding a much sharper and more accurate result.

## 3.4 ADAPTIVE ONE-STEP SAMPLING

Motivated by the path-crossing problem in multi-step sampling (Experiment 4.4), we find that single-step sampling via $z_0^{pred} = z_1 + \mathbf{v}_{1\to 0}$ yields comparable or superior performance. Nevertheless, we have observed that for a subset of challenging samples, this single-step process can yield blurry predictions. Based on Figure 3, these artifacts resemble an intermediate state along the flow trajectory that has not fully converged to the target. We hypothesize that this phenomenon is attributable to the model's underestimation of the magnitude of the predicted velocity vector $\mathbf{v}$. Consequently, the predicted state $z_0^{pred}$ fails to completely reach the ground-truth target $z_0^{truth}$, thereby introducing a residual offset between them. To rectify this issue, we propose a novel methodology termed Adaptive Single-Step Sampling.

To ensure precise alignment of $z_0^{\text{pred}}$ with the ground-truth distribution, AOS anchors the background regions of the predicted latent to a black reference latent $z_b = \Phi_{\text{encoder}}(I_b)$, where $I_b$ is a pure black image. Given the predicted velocity

$$\Delta z_{1\to 0}^{\text{pred}} = \mathbf{v}_{1\to 0}^{\text{pred}} \cdot \Delta t, \quad z_0^{\text{pred}} = z_1 + \Delta z_{1\to 0}^{\text{pred}}, \tag{5}$$

we first compute a candidate latent $z_0^{\text{pred}}$ from the single-step update. To identify reliable points for correction, we introduce the stable region index set

$$\mathcal{S} = \left\{ i \mid \left| z_b[i] - z_0^{\text{pred}}[i] \right| < \epsilon \right\}, \tag{6}$$

which collects the positions where the predicted latent is sufficiently close to the black reference latent. These points are assumed to correspond to background areas that should remain consistent across updates.

Within this stable region $\mathcal{S}$, we measure the average deviation between $z_0^{\text{pred}}$ and the reference latent relative to the predicted update magnitude. This ratio defines the adaptive scaling factor:

$$\gamma = \frac{\sum_{i \in \mathcal{S}} \left| z_b[i] - z_0^{\text{pred}}[i] \right|}{\sum_{i \in \mathcal{S}} \left| \Delta z_{1\to 0}^{\text{pred}}[i] \right|}, \qquad z_0^{\text{AOS}} = z_1 + \Delta z_{1\to 0}^{\text{pred}} \cdot (1 + \gamma). \tag{7}$$

By adaptively rescaling the update in proportion to the deviation observed in stable background regions, AOS compensates for prediction drift and enforces more consistent background alignment. This mechanism improves the overall trajectory of the rectified flow, leading to more accurate boundary localization and greater robustness against noisy predictions.

## 4 EXPERIMENTS

### 4.1 EXPERIMENTS SETTING

**Implementation details.** In our experiments, we adopt the Stable Diffusion v1.5 Rombach et al. (2022) checkpoint as the foundation for our Latent Diffusion Model (LDM), which utilizes the

Table 1: Performance comparison of text-based segmentation methods on the PhraseCut test set. Our approach outperforms all other compared methods in terms of mIoU.

| Method | mIoU | AP |
|---|---|---|
| RMI | 21.1 | - |
| Mask-RCNN Top | 39.4 | - |
| HulaNet | 41.3 | - |
| CLIPSeg (PC+) | 43.4 | 76.7 |
| CLIPSeg (PC, D=128) | 48.2 | 78.2 |
| RGBNet | 46.7 | 77.2 |
| LD-ZNet | 52.7 | **78.9** |
| RLFSeg(Ours) | **56.1** | 77.3 |

Figure 4: **Qualitative comparison with different methods.**

ViT-L/14 CLIP text encoder Radford et al. (2021) in a frozen state. During the training phase, we preserve the original parameters of the Stable Diffusion model and focus on fine-tuning only the LoRA Hu et al. (2022) layers, with a fixed rank of 64 applied throughout. The training is conducted on 8 NVIDIA A100 GPUs, each processing a batch size of 8, with the Adam optimizer and a base learning rate of 1e-4 for each mini-batch sample on each GPU.

**Dataset.** We follow LD-ZNet Pnvr et al. (2023) and evaluate on several benchmark datasets. Phrase-Cut Wu et al. (2020), the largest dataset for text-based image segmentation with 340K phrase–mask pairs, provides annotations for both stuff classes and multiple object instances. To further test generalization, we take the model trained on the PhraseCut training set and directly evaluate it on the referring expression segmentation benchmarks RefCOCO Kazemzadeh et al. (2014), Ref-COCO+ Kazemzadeh et al. (2014), and G-Ref Nagaraja et al. (2016). RefCOCO consists of short expressions (avg. 3.6 words) with at least two objects per image, while RefCOCO+ removes location words and focuses on appearance-based descriptions, making it more challenging. G-Ref contains longer expressions (avg. 8.4 words) with richer appearance and location details. We adopt the UNC partition for RefCOCO/RefCOCO+ and the UMD partition for G-Ref.

**Metrics.** Following LD-ZNet Pnvr et al. (2023), we report two evaluation metrics: the best mean Intersection-over-Union (mIoU) and the Average Precision (AP). The mIoU measures the overall pixel-level overlap between the predicted segmentation and the ground truth, providing a comprehensive assessment of segmentation accuracy. The AP evaluates the precision–recall trade-off across different thresholds, reflecting the model's ability to localize the regions referred to in text.

## 4.2 QUANTITATIVE EVALUATIONS

We quantitatively compare our method, RLFSeg, with state-of-the-art and baseline methods on four standard benchmarks. Our evaluation includes prominent approaches such as the VDPZhao et al. (2023), the ADDPPang et al. (2025), the LD-ZNet Pnvr et al. (2023), the CLIPSeg Lüddecke & Ecker (2022), and other established methods like HulaNet Wu et al. (2020) and RGBNet.

On the PhraseCut benchmark, our RLFSeg achieves the highest mIoU of 56.1 and a competitive AP of 77.3, surpassing previous methods such as CLIPSeg (48.2 mIoU, 78.2 AP)Lüddecke & Ecker (2022), RGBNet (46.7 mIoU, 77.2 AP), and LD-ZNet (52.7 mIoU, 78.9 AP), as detailed in Tab.1. These results highlight RLFSeg's ability to capture fine-grained semantics and generate precise segmentation masks directly from textual prompts. Unlike prior methods that rely on extra U-Net branches (e.g.,LD-ZNet) or handcrafted architectural designs (e.g., RGBNet), RLFSeg leverages rectified latent flows with adaptive refinement, achieving more accurate boundary alignment without additional architectural complexity.

As detailed in Tab. 2, RLFSeg also consistently outperforms existing methods on the more challenging referring expression benchmarks. For instance, on RefCOCO, RLFSeg obtains 42.5 mIoU and 50.9 AP, far exceeding the strongest baseline, LD-ZNet (41.0 mIoU, 17.2 AP). On RefCOCO+,

Table 2: Text-based segmentation results on RefCOCO, RefCOCO+, and G-Ref. We report mean IoU (mIoU) and Average Precision (AP) for each dataset. For fair comparison, all SD-based baseline models were pre-trained on the same dataset as Stable Diffusion v1.5.

| Method | RefCOCO | | RefCOCO+ | | G-Ref | |
|---|---|---|---|---|---|---|
| | mIoU | AP | mIoU | AP | mIoU | AP |
| *Zero-shot* | | | | | | |
| CLIPSeg (PC+) | 30.1 | 14.1 | 30.3 | 15.5 | 33.8 | 23.7 |
| RGBNet | 36.3 | 15.7 | 37.1 | 16.7 | 41.9 | 27.8 |
| LD-ZNet | 41.0 | 17.2 | 42.5 | 18.6 | 47.8 | 30.8 |
| RLFSeg(Ours) | **42.5** | **50.9** | **43.4** | **52.5** | **51.8** | **60.0** |
| *Fully Supervised* | | | | | | |
| VPD | 73.3 | - | 62.7 | - | 62.0 | - |
| ADDP | 69.1 | - | 57.6 | - | 59.0 | - |
| RLFSeg(Ours) | **75.3** | **81.7** | **66.0** | **71.8** | **67.8** | **74.1** |

Table 3: Segmentation results on the COCO-Stuff dataset. Our method shows a notable improvement in mIoU.

| Method | COCO-Stuff | |
|---|---|---|
| | mIoU | AP |
| SemFlow | 38.6 | - |
| **Ours** | **39.7** | **59.0** |

Table 4: Inference time comparison. The time is measured in seconds per image, averaged over 100 runs on a single NVIDIA RTX 3090 GPU.

| Method | Inference Time (s) |
|---|---|
| LD-ZNet | **0.17** |
| SemFlow | 1.09 |
| **Ours** | 0.19 |

RLFSeg reaches 43.4 mIoU and 52.5 AP, again outperforming LD-ZNet (42.5 mIoU, 18.6 AP). The advantage is most pronounced on G-Ref, where RLFSeg achieves 51.8 mIoU and 60.0 AP, compared to 47.8 mIoU and 30.8 AP of LD-ZNet. These consistent improvements across diverse datasets highlight the strong zero-shot generalization ability of our framework in handling complex queries and challenging object boundaries.

We directly compare our method with SemFlow to highlight key performance and efficiency differences. As shown in Table 3, our model achieves 39.7 mIoU on COCO-Stuff in a zero-shot setting, surpassing SemFlow's fully-trained result of 38.6 mIoU. This demonstrates the superior effectiveness of our architecture, which is optimized for discriminative precision. Furthermore, Table 4 reveals a significant efficiency gap: our model (0.19s) is approximately 5.7x faster than SemFlow (1.09s). This combination of higher zero-shot accuracy and substantially lower computational cost clearly establishes our method as a more practical and effective solution.

In summary, across all benchmarks, RLFSeg attains the best performance among diffusion-based segmentation approaches. By directly modeling latent transformations with rectified flows and incorporating refined supervision, our framework effectively narrows the generative–discriminative gap, producing segmentation masks that are both semantically consistent and boundary-accurate, while remaining efficient and robust to noisy annotations.

### 4.3 ABLATION STUDY

We demonstrate the effectiveness of each component of RLFSeg through ablation studies, with the results presented in Tab. 5. Compared to the baseline model (trained solely with the RF strategy), which achieves 55.3 mIoU, introducing the RDS strategy during training refines the learning objective, leading to a more precise mapping and an improved mIoU of 55.8. Furthermore, applying AOS during the inference stage further refines the latent features predicted by RLFSeg, raising the mIoU to 56.1. Both strategies independently improve the performance of the RF model, and when combined, they provide cumulative performance gains.

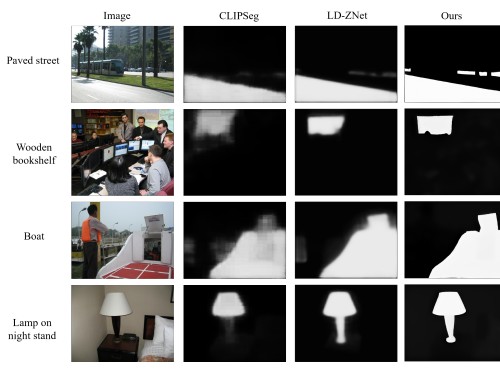 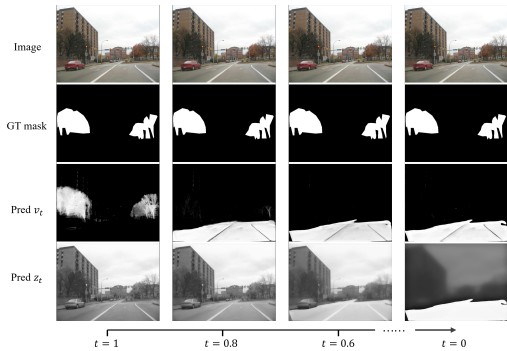

Figure 5: **Qualitative comparison of mask boundaries.** Our proposed method generates significantly sharper and better-defined edges compared to competing approaches.

Figure 6: **Visualization of the path-crossing issues.** This figure shows that at $t = 0.8$, the predicted direction of v changes drastically compared to its direction at $t = 1.0$

We conducted an ablation study to determine the optimal LoRA rank for our fine-tuning process. As detailed in Tab. 6, performance peaks at a rank of 64 (56.1 mIoU), with higher values yielding no significant improvement. Most strikingly, abandoning LoRA for full-parameter fine-tuning causes a severe performance collapse to 51.9 mIoU. This result strongly suggests that full fine-tuning catastrophically damages the model's essential pre-trained priors, underscoring that a parameter-efficient approach like LoRA is crucial for effectively leveraging the diffusion model for our task.

Table 5: Comparison of performance with different strategies. RDS denotes the Refinement with Dynamic Selection, and AOS denotes Adaptive One-Step Sampling.

| RDS | AOS | mIoU | AP |
|-----|-----|------|------|
| ✗ | ✗ | 55.3 | 76.7 |
| ✗ | ✓ | 55.4 | **78.5** |
| ✓ | ✗ | 55.8 | 76.9 |
| ✓ | ✓ | **56.1** | 77.3 |

Table 6: Ablation study on the effect of LoRA rank on text-based segmentation performance. "Full Param" indicates the baseline model without LoRA.

| Rank | mIoU | AP |
|------|------|------|
| 16 | 55.3 | 76.7 |
| 32 | 55.7 | 76.3 |
| 64 | 56.1 | 77.0 |
| 128 | 56.1 | 76.6 |
| 256 | 56.0 | 77.5 |
| Full Params | 51.9 | 73.1 |

### 4.4 QUALITATIVE ANALYSES

**Precision in Detail.** Figure 4 showcases several examples where our method achieves more precise segmentation results. For instance, in the "short hair" case, our approach demonstrates a more accurate grasp of the hair's contours compared to other methods. In the "tree standing alongside road" example, RLFSeg successfully filters out the empty spaces between the tree branches. While ClipSeg can also filter these hollow regions to some extent, it introduces other artifacts. In contrast, LD-ZNet recognizes the overall tree structure well but fails to handle the hollow areas within the branches.

**Well-defined boundaries.** As can be seen in Figure 5 Unlike most conventional segmentation methods, the masks generated by RLFSeg exhibit remarkably sharp and well-defined boundaries, a characteristic reminiscent of generative models' proficiency in synthesizing high-frequency details. We attribute this advantage to our training strategy of learning the ground truth distribution in the latent space, which allows our model to distinguish itself from numerous other segmentation approaches.

**Path-crossing issues.** When utilizing Rectified Flow, the significant distributional overlap between the source $z_0$ (RGB color images) and the target $z_1$ (grayscale masks) can lead to path-crossing issues which is a phenomenon where a single initial state's generative trajectory bifurcates, leading to multiple distinct terminal points, despite the deterministic nature of the semantic segmentation task where a unique mapping from image to mask should exist. We found this problem to be particularly

acute in the early stages of the trajectory, specifically between timesteps $t = 0.9$ and $t = 0.7$. During this critical phase, the model is determining its initial direction, making it highly susceptible to interference from the overlapping distributions. This can cause incorrect predictions, such as confusing the white foreground of the mask with bright white regions in the original image. Consequently, increasing the number of sampling steps can sometimes degrade performance. Figure 6. presents a particularly extreme case of this failure mode; however, it is important to note that most cases are not this severe.

## 5 CONCLUSION

In our work, we have proposed RLFSeg, a method that integrates the traditional task of semantic segmentation with flow matching. Within this framework, the segmentation task is seamlessly fused with the Latent Diffusion Model (LDM) architecture, in contrast to previous works that often rigidly employed diffusion models as internal feature extractors. This approach allows us to bypass the problem of timestep selection for the image segmentation task and avoids the dependency on random sampling noise inherent to original diffusion models, leading to a more streamlined and elegant process design. Through comprehensive experiments, we have demonstrated that our method not only achieves strong results on in-distribution test sets but also exhibits superior generalization capabilities compared to prior works in this research area.

**Reproducibility Statement.** To ensure the full reproducibility of our findings and to facilitate future research in this area, we have made comprehensive efforts to document our work. Section A.3 detail the availability of our data, experimental setup, and all necessary parameters.

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

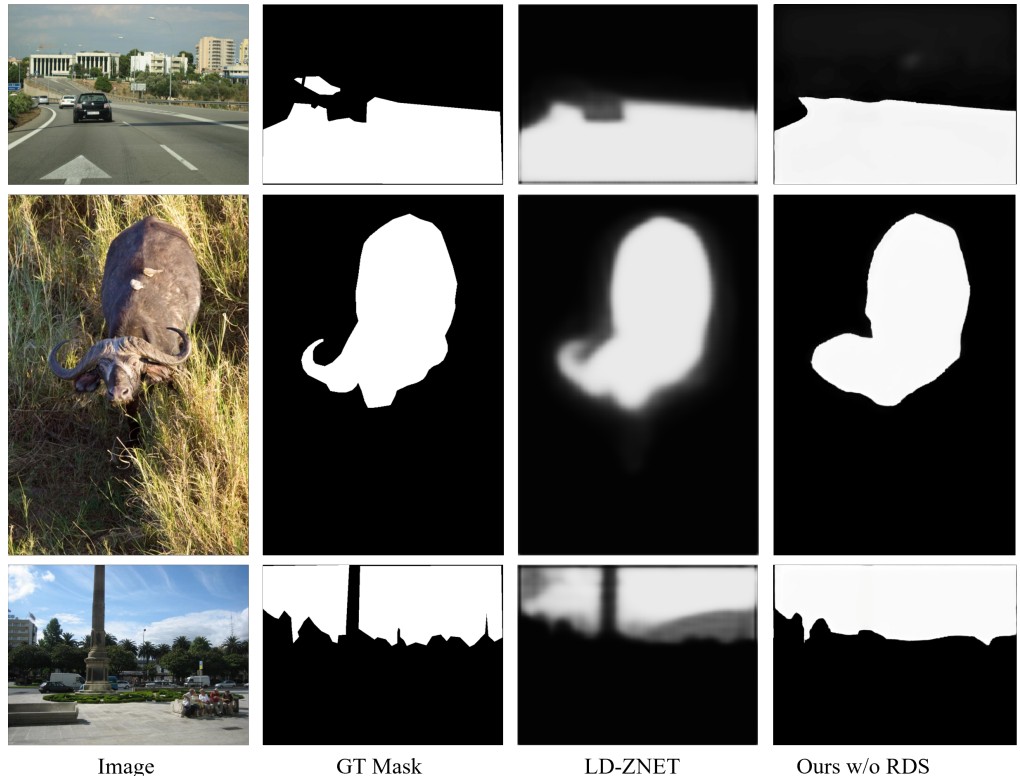

| Image | GT Mask | LD-ZNET | Ours w/o RDS |

Figure 7: **Visualization of results without RDS.** The Rectified Flow training can cause the model to predict polygon-like masks in some cases, which reduces segmentation accuracy.

# A APPENDIX

You may include other additional sections here.

## A.1 SIDE EFFECTS OF POLYGON-BASED ANNOTATIONS

Training with polygon-based annotations causes the model to mimic the "jagged" or "low-fidelity" style inherent in such labels. This issue is significantly less severe in conventional semantic segmentation frameworks that use cross-entropy loss(e.g., ldznet), as they are less sensitive to precise boundary details. Figure 7 showcases several illustrative examples from the test set where this stylistic overfitting is apparent. While not all predictions degrade to this level of quality, these cases demonstrate a clear bias the model acquires, leading to a propensity for producing coarse or unnatural boundaries.

## A.2 HYBRID STEP SAMPLING

While adaptive scaling refines the single-step update locally, multi-step sampling provides a robust estimate of the overall step length, despite gradually degrading directional accuracy. To leverage both strengths, we propose Hybrid Step Sampling(HSS), which rescales the single-step update according to the cumulative step length from a multi-step trajectory.

For a $K$-step multi-step process, let each step predict velocity $\mathbf{v}_k$ with integration time $\Delta t_k$. In our experiments, we use a 5-step process (i.e., K=5). The multi-step trajectory produces a reference latent

$$z_0^{\mathrm{multi}} = \mathrm{MultiStepSample}\big(z_1, \{\mathbf{v}_k, \Delta t_k\}_{k=1}^K\big). \tag{8}$$

Define the stable-region index set

$$\mathcal{S} = \big\{i \mid \big|z_0^{\mathrm{multi}}[i] - (z_1 + \Delta z_{1\to0}^{\mathrm{pred}})[i]\big| < \epsilon\big\}, \tag{9}$$

Table 7: Performance comparison of RLFSeg with AOS and HSS. The table indicates that both strategies contribute comparable performance gains.

| method | mIoU | AP |
|---|---|---|
| RLFSeg with AOS | 56.1 | 77.2 |
| RLFSeg with HSS | 56.1 | 77.3 |

i.e., the set of positions where the single-step and multi-step predictions nearly agree.

The single-step update $\Delta z_{1\to 0}^{\text{pred}} = \mathbf{v}_{1\to 0}^{\text{pred}} \cdot \Delta t_1$ is then adaptively rescaled using only pixels in $\mathcal{S}$:

$$\gamma = \frac{\sum_{i\in\mathcal{S}} \left| z_0^{\text{multi}}[i] - (z_1 + \Delta z_{1\to 0}^{\text{pred}})[i] \right|}{\sum_{i\in\mathcal{S}} \left| \Delta z_{1\to 0}^{\text{pred}}[i] \right|}, \qquad z_0^{\text{HSS}} = z_1 + \Delta z_{1\to 0}^{\text{pred}} \cdot (1 + \gamma), \qquad (10)$$

where $|\mathcal{S}|$ denotes the cardinality of $\mathcal{S}$. (If $|\mathcal{S}| = 0$ we fall back to $\gamma = 0$.)

This unified approach preserves the accurate directional flow from the single-step prediction while incorporating the robust total step length from multi-step sampling, improving boundary precision and overall mask quality.As shown in Table 7, while this method achieves comparable performance to AOS, it incurs a longer inference time due to its requirement for multi-step sampling. Consequently, we did not incorporate this method into the final RLFSeg model.

### A.3 EXPERIMENTAL DETAILS

This section provides the technical details required to reproduce the core experiments of this paper. It covers the hardware and software environment, data processing pipeline, model configuration, and the specific steps for training and evaluation.

#### A.3.1 ENVIRONMENT SETUP

To reproduce our results, the first step is to set up the appropriate hardware and software environment. We recommend creating a dedicated Python environment. The key dependencies are listed below. For a complete list, please refer to `requirements.txt`.

- **OS:** Debian GNU/Linux 12
- **CUDA Version:** 12.4
- **Python Version:** 3.11.2
- **Core Framework:** PyTorch 2.5.1
- **Key Libraries:** transformers 4.52.4, diffusers 0.33.1, accelerate 1.7.0

#### A.3.2 DATASET AND PREPROCESSING

**Datasets.** Our experiments are conducted on several public benchmarks. We train our model exclusively on the **PhraseCut** training set and evaluate its performance on the validation/test sets of **PhraseCut**, **RefCOCO**, **RefCOCO+**, and **G-Ref**. The latter three are used to assess the model's zero-shot generalization ability.

- **PhraseCut** Wu et al. (2020): This is the primary dataset for both training and evaluation. It is the largest of its kind, with 340K phrase-mask pairs covering a wide range of object instances and stuff classes. We use its official training set for training and its validation/test set for evaluation.
  - *Source:* `https://github.com/ChenyunWu/PhraseCutDataset`
- **Zero-Shot Evaluation Datasets:** To test generalization to unseen datasets, we directly evaluate our PhraseCut-trained model on the following benchmarks using their standard data splits:
  - **RefCOCO** & **RefCOCO+** Kazemzadeh et al. (2014): We adopt the UNC partition. RefCOCO+ is particularly challenging as it omits locational words.

- **G-Ref** Nagaraja et al. (2016): We use the UMD partition. This dataset features longer and more descriptive expressions.
- *Source (for all three):* https://github.com/lichengunc/refer

**PreProcessing.** To enhance development efficiency and accelerate data I/O, we consolidated all datasets into the HDF5 format. In this structure, each sample is stored as a top-level HDF5 **Group** named after its unique image_id. This group contains two components: (1) a **Dataset** with the fixed key 'image', which stores the image NumPy array, and (2) a nested **Group** with the fixed key 'mask'. This 'mask' group, in turn, holds the individual mask **Datasets**, where each dataset is keyed by its corresponding text caption and stores the mask's NumPy array.

### A.3.3    MODEL ARCHITECTURE AND TRAINING CONFIGURATION

**Model Architecture.** We adopt the pretrained **Stable Diffusion v1.5** Rombach et al. (2022) as our foundational model. For all experiments, we utilize the specific checkpoint from the Hugging Face Hub[1] and make no modifications to its standard U-Net and VAE architecture.

**Training Hyperparameters** The key hyperparameters for our training process are summarized in Table 8. We employ a parameter-efficient fine-tuning strategy using Low-Rank Adaptation (LoRA) on the Stable Diffusion v1.5 model, keeping the original U-Net weights frozen.

Table 8: Key training hyperparameters.

| Hyperparameter | Value |
|---|---|
| Fine-tuning Method | LoRA (Low-Rank Adaptation) |
| LoRA Rank | 64 |
| Optimizer | AdamW |
| Learning Rate | 1e-4 |
| Batch Size (per GPU) | 8 |
| Training Epochs | 10 |
| Image Resolution | $512 \times 512$ |
| Random Seed | 42 |

### A.4    ABLATION STUDY ON THE NUMBER OF SAMPLING STEPS

To investigate the impact of the number of sampling steps on performance, we conducted an ablation study, with results presented in Table 9. We evaluated our model using 1, 2, 5, and 15 steps on four different benchmarks. For a fair comparison across all runs, the Add-on-Saliency (AOS) module was disabled.

Table 9: Ablation study on the number of sampling steps. We report mIoU and AP on four datasets. For a fair comparison, Add-on-Saliency (AOS) was not used in any experiment. A single sampling step consistently yields the best performance across all benchmarks.

| Steps | PhraseCut | | RefCOCO | | RefCOCO+ | | G-Ref | |
|---|---|---|---|---|---|---|---|---|
| | mIoU | AP | mIoU | AP | mIoU | AP | mIoU | AP |
| 1 | **55.8** | **77.1** | **42.2** | **49.3** | **43.1** | **51.0** | **51.6** | **59.3** |
| 2 | 47.2 | 72.7 | 37.0 | 44.2 | 36.7 | 45.2 | 44.2 | 53.8 |
| 5 | 46.0 | 71.1 | 36.4 | 43.5 | 36.0 | 44.3 | 43.6 | 52.6 |
| 15 | 44.5 | 69.7 | 35.6 | 43.0 | 35.2 | 43.8 | 42.7 | 51.8 |

The results clearly indicate that a single sampling step achieves the best performance across all datasets. For instance, on PhraseCut, the mIoU drops from **55.8** with one step to 47.2 with two steps.

---

[1]Model ID: sd-legacy/stable-diffusion-v1-5. Available at: https://huggingface.co/stable-diffusion-v1-5/stable-diffusion-v1-5

A similar trend is observed on the other datasets, with performance consistently degrading as the number of sampling steps increases. This finding supports our hypothesis that for a discriminative task like segmentation, a direct, one-step mapping is more effective than an iterative refinement process, which may introduce noise or deviate from the optimal solution. Consequently, we use a single sampling step for all other experiments in this paper.

