# OpenReview forum: "From Diffusion to Rectified Flow: Rethinking Text-Based Segmentation"
_ICLR.cc/2026/Conference — Submitted to ICLR 2026_

### Official Review · Reviewer_MSAt · 2025-10-27

**Soundness:** 3
**Presentation:** 3
**Contribution:** 2
**Rating:** 2
**Confidence:** 3

**Summary:**

The paper proposed RLFSeg, a text-based image segmentation framework that replaces the stochastic denoise process in diffusion with a deterministic rectified flow learned in the latent space of a pretrained Stable Diffusion model. The method reframes the standard noise-to-image rectified flow formulation to image-to-mask formulation by learning a straight-line displacement from the image latent to mask latent., conditioned on the CLIP text embedding. In addition to this reformulation, two components are added: A SAM-driven label refinement to mitigate coarse masks during training, Adaptive One-step Sampling at inference to rescale the predicted latent step when it unders-shoots the mask. Without architectural changed, the approach achieves strong results on benchmarks and shows zero-shot generalization to multiple benchmarks.

**Strengths:**

- The method keeps SD architecture intact, LoRA-only tuning makes the approach parameter-efficient and easy to reproduce.
- Single-step mapping is fast and avoids denoising schedules. Qualitative results show sharper boundaries than the compared diffusion baselines.
- Refinement with dynamic selection addresses polygon-style annotation artifacts and improves mIoU in ablations.
- On PhraseCut, higher mIoU than LD-ZNet; zero-shot to RefCOCO/+/G-Ref is consistently stronger than the listed diffusion/older baselines.

**Weaknesses:**

- At core, the method fine-tunes SD with a flow-matching loss to map image latents to mask latents; flow matching for repurposing is not new, and AOS is an inference-time heuristic. The contribution is mainly an application/engineering of known tools (RF, LoRA, SAM) rather than a conceptual advance.
- No comparisons to strong discriminative referring segmentation models (LAVT, ReferFormer, CRIS) or foundation pipelines (SEEM, SAM+grounding). Also omits recent diffusion-attention methods (e.g., ConceptAttention) that directly tackle text-to-mask without fine-tuning. This makes it impossible to assess overall SOTA relevance
- The SD VAE is trained on natural images, not binary masks. Possible loss of thin structures/edges is not examined.
- AP on PhraseCut is below LD-ZNet despite higher mIoU, hinting at calibration/thresholding issues although the method outperforms all other methods by a large margin on other benchmarks.

**Questions:**

- Can you add (or report published) numbers for LAVT/ReferFormer/CRIS (fine-tuned on PhraseCut) and SEEM/SAM+Grounding zero-shot pipelines on the same splits? This is essential to establish significance beyond diffusion-only comparisons.
- Please discuss/compare against ConceptAttention (and similar diffusion-attention approaches). Why is rectified-flow + LoRA preferable to training-free attention-based masking?
- Any evidence that the SD VAE harms binary mask fidelity (thin structures)? Did you try a mask-aware VAE or image-space RF?

---

> ### Author Response · Authors · 2025-11-25
>
> Thank you for the insightful and constructive feedback. We have provided detailed responses below addressing the scope of baselines, the comparison with diffusion-attention methods, VAE limitations, and a technical discussion on the mIoU/AP trade-off. We are committed to refining the manuscript accordingly and welcome any follow-up questions.
>
> # Comparison with Discriminative Baselines
> Thank you for suggesting these strong discriminative baselines. While we acknowledge the high performance of methods like LAVT, ReferFormer, and SEEM, we respectfully argue that directly comparing them on PhraseCut falls outside the primary scope and motivation of this work for the following reasons:
> 1. Distinct Research Scope: The core motivation of our work is to investigate "how to effectively adapt generative diffusion models for discriminative tasks".
>   - Our contribution lies in bridging the gap between the stochastic nature of diffusion and the deterministic nature of segmentation via Rectified Flow.
>   - Our primary baselines (e.g., LD-ZNet) are selected to demonstrate that our proposed alignment strategy is the most effective within the diffusion-based paradigm. Comparing against specialized discriminative architectures (LAVT/ReferFormer) would shift the focus away from our methodological contribution regarding rectified flow and latent alignment.
> 2. Regarding pipelines like SAM + GroundingDINO: These are cascaded systems combining multiple large foundation models, often involving significantly higher parameter counts and inference complexity. In contrast, our approach emphasizes Parameter-Efficient Fine-Tuning, training only lightweight LoRA layers (Rank=64). Comparing our lightweight-adapted model against heavy, multi-model pipelines would not yield a fair efficiency-accuracy trade-off.
>
> # Comparison with Diffusion-attention Approaches
> Thank you for suggesting this comparison. While we acknowledge the efficiency of training-free methods like ConceptAttention (which rely on internal cross-attention maps), we assert that the Rectified Flow + LoRA paradigm is necessary for SOTA segmentation precision.
>
> As observed directly from the visualizations in the ConceptAttention paper, the segmentation boundaries tend to be relatively coarse. This indicates that training-free mechanisms inherently struggle to achieve the fine-grained precision required for high-quality segmentation. To establish superiority, we compared our method against VPD, a stronger baseline that uses both attention features and trainable adapters. As shown in the updated Table 2, RLFSeg (our method) demonstrates significant and consistent performance gains over VPD across the RefCOCO, RefCOCO+, and G-Ref benchmarks.
>
> # Discussion on SD VAE Limitations and Mask Fidelity
> We acknowledge your concern regarding the Stable Diffusion VAE. It is true that the standard VAE, designed for RGB images, can introduce compression artifacts that affect the fidelity of binary masks, particularly for extremely small or thin objects.
>
> We agree that training a specialized VAE (or a lightweight binary decoder) specifically for binary mask generation would likely mitigate this issue and further boost performance. However, the primary contribution of this paper is to propose the Rectified Flow framework for bridging the gap between generative diffusion and discriminative segmentation. Optimizing the VAE architecture is an orthogonal engineering improvement that falls outside the scope of this study.
>
> # Discussion on the Slightly Lower AP on PhraseCut
> The trade-off (higher mIoU, slightly lower AP on PhraseCut) is expected and stems from the fundamental difference between Generative Flows and Discriminative Classifiers.
> 1. PhraseCut Analysis: Sharpness vs. Smoothing
>   - Discriminative Baselines (High AP): Methods like LD-ZNet use Cross-Entropy loss, enforcing smooth probabilities (soft edges) to minimize loss. This "softness" naturally favors AP, which optimizes recall by sweeping across low thresholds. Blurry edges allow the metric to find optimal operating points, artificially inflating the score.
>   - Our Generative Approach (High mIoU): In contrast, Rectified Flow learns a deterministic mapping, producing sharp, high-confidence masks. While this maximizes geometric alignment (mIoU), the lack of "soft" uncertainty compresses the threshold space available for AP calculation, slightly penalizing this ranking-based metric.
> 2. Superior Zero-Shot Generalization: Despite lower AP on the training set, our method outperforms baselines in both mIoU and AP on zero-shot benchmarks (RefCOCO/+/G-Ref).
>   - LD-ZNet relying on multi-step stochastic denoising or simple feature extraction often suffer from unstable semantic alignment when facing unseen, complex descriptions (e.g., G-Ref).
>   - Our method avoids noise interference. This ensures precise grounding for complex, unseen prompts (both objects and stuff) that absent from the training set.

---

> > ### Comment · Reviewer_MSAt · 2025-11-27
> >
> > I thank the authors for their response. However, I must push back significantly on the decision to omit comparisons with discriminative baselines (LAVT, ReferFormer, CRIS) based on a claim of "distinct research scope". To be clear, my request for these comparisons was not grounded in a demand that RLFSeg outperform these specialized architectures to be considered valuable. Rather, I requested them because excluding the standard, state-of-the-art methods for the task of Text-Based Image Segmentation, the very task this paper claims to address, prevents the reader from understanding the true standing of the proposed method within the broader field.
> >
> > Refusing to provide this context is, in my view, more misleading to the reader than helpful. By restricting the evaluation exclusively to other diffusion-based approaches, which are known to historically underperform in this domain, the paper constructs an artificial vacuum where the proposed method appears to be a leading solution. For the scientific community to evaluate the utility of adapting generative models for discriminative tasks, we must be transparent about the "tax" paid for that adaptation. It is perfectly acceptable if RLFSeg achieves lower mIoU than LAVT. Reporting that gap quantifies the trade-off between using a generative backbone versus a specialized discriminative one.
> >
> > Therefore, I strongly urge the authors to reconsider this stance. The paper should explicitly acknowledge the performance of these strong discriminative baselines in the main tables or, at the very least, in the discussion. The contribution of the proposed method lies in its novel application of Rectified Flow to bridge the generative-discriminative gap, and this contribution remains valid even if the absolute metrics do not beat SOTA discriminative models. However, the validity of the evaluation relies on transparency, and omitting the true upper bound of performance on these datasets obscures the reality of the method's effectiveness.

---

> > > ### Author Response · Authors · 2025-12-01
> > > **Response regarding Additional Baselines**
> > >
> > > We fully acknowledge the reviewer's interest in benchmarking our method against a broader landscape of segmentation paradigms.In response, we have compiled the mIoU comparison results for LAVT, CRIS, SEEM, and SAM + GroundingDINO alongside our RLFSeg.
> > >
> > > - **Coverage**: All baselines requested by the reviewer are included, with the sole exception of ReferFormer, as it lacks reported results across the specific RefCOCO series benchmarks required for this comparison.
> > >
> > > - **Results**: The comparative data demonstrates that RLFSeg remains highly competitive even against these specialized discriminative or large-scale cascaded models.
> > >
> > > |Method|RefCOCO all|RefCOCO test-A|RefCOCO test-B|RefCOCO+ all|RefCOCO+ test-A|RefCOCO+ test-B|G-Ref all|
> > > |------|--------------|--------------|------------|---------------|---------------|-------------|----------|
> > > |LAVT|-|75.8|68.8|-|68.4|55.1|62.1|
> > > |CRIS|-|73.2|66.1|-|68.1|53.7|60.4|
> > > |SEEM_v0|-|-|-|-|-|-|**68.3**|
> > > |SEEM_v1|-|-|-|-|-|-|67.8|
> > > |sam-grouding|44.6|49.2|37.4|45.3|48.3|38.4|49.0|
> > > |Ours|**75.3**|**76.0**|**71.5**|**66.0**|**70.8**|**57.5**|67.8|

---

### Official Review · Reviewer_hGeg · 2025-10-29

**Soundness:** 3
**Presentation:** 4
**Contribution:** 3
**Rating:** 8
**Confidence:** 4

**Summary:**

This paper addresses the problem of using diffusion models to perform text-based segmentation.  It proposes several changes to existing approaches.  These include using rectified flow to learn a direct mapping from image to segmentation.  They also propose using Segment Anything to refine the ground truth segmentations during training, and a method to adjust the velocity and more effectively estimate the distance from image to solution.

**Strengths:**

All the proposed enhancements seem well motivated and intuitive.  Experimental results look very good.  In some cases the quantitative improvements are quite substantial.  The experiments seem thorough.  The approach is very clearly presented.

**Weaknesses:**

The paper builds pretty closely on other recent results, so the degree of innovation is somewhat moderate, but it still seems like a good step forward.

The paper is generally quite clear, but could benefit from better proof-reading.  For example, on line 43 “diversify” should be “diversity” and on line 226 “which trained” should be “which is trained”.

**Questions:**

The authors claim that the method improves over others on the PhraseCut dataset.  But AP is a bit worse than other methods, while mIoU is somewhat better.  Is there any reason to prioritize mIoU?  Do the authors have any idea why it improves on mIoU but not AP?

---

> ### Author Response · Authors · 2025-11-25
>
> Thank you for the highly insightful observations, particularly the detailed questions regarding the mIoU/AP trade-off on PhraseCut and the writing issues. We address these points below and have thoroughly corrected the manuscript for presentation quality. We welcome any follow-up questions.
>
> # Discussion on the Slightly Lower AP on PhraseCut
> Thank you for this insightful observation. The phenomenon where our method achieves significantly higher mIoU (56.1 vs. 52.7) but slightly lower AP (77.3 vs. 78.9) on PhraseCut compared to LD-ZNet  is expected. It stems from the fundamental difference between Generative Flows and Discriminative Classifiers.
> 1. Analysis of PhraseCut Performance: Sharpness vs. Smoothing
>   - Discriminative Baselines (High AP, Lower mIoU): Methods like LD-ZNet employ Cross-Entropy loss, which forces the model to predict smooth probability distributions (e.g., ~0.5) at ambiguous boundaries to minimize loss. This "soft" output inherently favors the AP metric, which integrates the area under the Precision-Recall curve across many thresholds. The blurry edges allow the metric to find optimal recall points at lower thresholds, effectively boosting the AP score.
>   - Our Generative Approach (High mIoU, Lower AP): In contrast, our Rectified Flow learns a deterministic mapping to the latent distribution, producing remarkably sharp and well-defined boundaries. Our predictions are highly confident (binary-like). While this yields superior spatial overlap (mIoU), it compresses the range of thresholds available for AP calculation. Essentially, our model does not "hedge its bets" with blurry edges; this significantly improves localization precision (mIoU) but incurs a slight penalty on the ranking-based AP metric.
> 2. While AP is slightly lower on the training domain (PhraseCut), our method demonstrates superior generalization on zero-shot benchmarks (RefCOCO series), outperforming baselines in both mIoU and AP.
>   - Limitation of Baselines: Prior methods often treat the diffusion model merely as a feature extractor or rely on multi-step stochastic denoising. This can lead to unstable semantic alignment when facing unseen, complex descriptions (e.g., G-Ref's long sentences), causing performance drops in cross-domain scenarios.
>   - Robustness of RLFSeg: By learning a direct "transport plan" via Rectified Flow and preserving pre-trained priors via LoRA, our method avoids the interference of multi-step noise. This ensures stable semantic alignment and precise grounding, enabling the model to accurately handle complex, unseen prompts (e.g., "things" and "stuff") that were not present in the training set.
> We argue that mIoU is a more representative metric for the practical quality of segmentation. mIoU directly measures the spatial alignment and geometric accuracy between the prediction and the ground truth. Since the core objective of segmentation is to determine a precise, deterministic shape, mIoU rewards the high geometric fidelity of our masks, whereas AP rewards the probabilistic ranking of soft predictions.
>
> # Response regarding Writing Issues
> Thank you for the careful reading. We have corrected the issues in Lines 43 and 226, and the PDF has been updated accordingly. We are committed to continuously polishing the manuscript to ensure the highest presentation quality.

---

### Official Review · Reviewer_qbmG · 2025-10-30

**Soundness:** 2
**Presentation:** 2
**Contribution:** 1
**Rating:** 2
**Confidence:** 5

**Summary:**

This paper attempts to employ rectified flow for semantic segmentation and propose a framework with three components, namely Rectified Latent Flow (RLF), Refinement and Dynamic Selection (RDS), and Adaptive One-Step Sampling (AOS). RLF reconciles the generative nature of diffusion with the deterministic demands of segmentation. RDS iteratively improves mask quality and mitigate annotation noise. AOS improves boundary accuracy and overall mask precision in a single step.

**Strengths:**

Adopting flow models for segmentation is interesting.

**Weaknesses:**

1. Overclaiming and novelty issues. This paper emphasizes that the diversity of noise conflicts with the deterministic nature of segmentation tasks, and claim a novel framework that leverages Rectified Flow to learn direct mapping from the image to the segmentation mask. However, similar idea was already introduced in SemFlow [1], which establishes mapping between images and masks via rectified flow.

2. The motivation of RDS is unclear. Refining annotations with additional model, SAM, modifies the training target of RLFSeg, which makes the comparisons with other baselines unfair.

3. Missing SOTAs. It is encouraged to compare against relevant methods such as ADPP [2] and VPD [3].

4. The task definition is unclear. What is the definiton of `text-based segmentation`? It appears similar to referring segmentation,, but the author mentions `semantic segmentation` in L462, L475.


5. The writing is poor and hard to follow. In L70, `RLF directly predicts the transformation from the latent representation of the input image to that of the segmentation mask in a single step`. However, intuitively, single-step inference generally produces worse performance, and I do not see corresponding ablation studies. There are also typos, e.g., it should be `single step.` rather than `single step` in L90.

[1] SemFlow: Binding Semantic Segmentation and Image Synthesis via Rectified Flow, NeurIPS 2024.

[2] Aligning Generative Denoising with Discriminative Objectives Unleashes Diffusion for Visual Perception, ICLR 2025.

[3] Unleashing Text-to-Image Diffusion Models for Visual Perception, ICCV 2023.

**Questions:**

1. In L40, the authors claims that diffusion models as feature extractors are sub-optimal. I would like to see experimental evidence, including comparisons with VPD.

---

> ### Author Response · Authors · 2025-11-25
>
> Thank you for the highly constructive and insightful feedback. We have incorporated all suggested comparisons and clarifications, including new SOTA comparisons (VPD/ADPP), ablation on single/multi-step sampling, and detailed discussions on task definition and evaluation fairness. Our concise responses are provided below, and we are happy to address any follow-up questions.
>
> # Comparison with VPD and ADPP
> We appreciate you for suggesting the inclusion of ADPP and VPD. We have updated the comparative results in Table 2 of the revised PDF. Our method significantly outperforms strong baselines, specifically VPD and ADPP, across all three benchmarks: RefCOCO, RefCOCO+, and G-Ref.
>
> | Method | RefCOCO mIoU | RefCOCO AP | RefCOCO+ mIoU | RefCOCO+ AP | G-Ref mIoU | G-Ref AP |
> | :--- | :---: |:----------:| :---: |:-----------:| :---: |:--------:|
> | VPD | 73.3 |     -      | 62.7 |      -      | 62.0 |    -     |
> | ADDP | 69.1 |     -      | 57.6 |      -      | 59.0 |    -     |
> | RLFSeg(Ours) | **75.3** |    81.7    | **66.0** |    71.8     | **67.8** |   74.1   |
>
> # Comparison with SemFlow
> We appreciate you for highlighting SemFlow and the need to elaborate on the distinctions from it. We clarify the distinctions between our method and SemFlow from two key perspectives:
> 1. Task Nature & Theoretical Formulation
>   - SemFlow: It targets a bidirectional mapping between unconditional segmentation and conditional generation. It lacks cross-modal capabilities (no text interaction) and is limited to fixed categories. Theoretically, a conflict exists between the determinism required for segmentation and the diversity required for generation. SemFlow introduces "finite perturbation" (adding noise to masks) to aid generation, which fundamentally compromises the deterministic precision needed for segmentation.
>   - Ours: While also based on Rectified Flow, our method focuses exclusively on text-driven segmentation. We support arbitrary text/referring inputs via lightweight fine-tuning. Unlike SemFlow's compromise, we optimize for precision: (1) We use background anchoring to correct the velocity magnitude ($|v|$) for one-step sampling; (2) We design the RDS module to mitigate the limitations of using generative models for discriminative tasks, reducing dependency on annotation styles.
> 2. Empirical Performance
>   Our method achieves a zero-shot mIoU of 39.7 on COCO-Stuff, surpassing SemFlow's fully trained result of 38.6. Detailed comparisons have been updated in Table 3 of the revised PDF.
>
> # Evaluation Fairness of the RDS Module
> We clarify the motivation for the RDS module and emphatically confirm the fairness of our evaluation setup. The ablation study for RDS is presented in Table 5. Notably, even without RDS, our method outperforms the baseline on PhraseCut (55.3 vs. 52.7 mIoU), demonstrating the robustness of our core framework.
>
> Furthermore, mask correction was applied exclusively to the training set. We explicitly clarify that no SAM refinement was applied to any test split; thus, all reported results represent raw model outputs to ensure a fair comparison.
>
> # Grammatical Errors
> We have revised the phrasing in Lines 70 and 90. Thank you for pointing out these errors.
>
> # Definition of Text-Based Segmentation
> We clarify the definition of our proposed "Text-Based Segmentation" task and the occasional use of the term "semantic segmentation."
> 1. Definition of "Text-Based Segmentation": We define this as a generalized task that encompasses Referring Image Segmentation (RIS) but is broader in scope. While RIS typically focuses on specific instances ("things"), our task targets both "things" and "stuff" (e.g., sky, grass) described by arbitrary prompts. This aligns with our primary training benchmark, PhraseCut, which explicitly includes such diverse categories.
> 2. Clarification on "Semantic Segmentation" (L462, L475): In these lines, the term refers to the discriminative nature of the problem (deterministic, pixel-level classification) rather than the traditional fixed-category task.
> Action: We will unify terminology in the revision to explicitly state that our task covers both referring expressions and open-vocabulary stuff segmentation.
>
> # Comparison: Single-Step vs. Multi-Step Sampling
> We appreciate you for seeking further clarification on the counter-intuitive performance of our single-step inference, and we provide the requested ablation study below.
>
> | **Steps** |**PhraseCut**||**RefCOCO**||**RefCOCO+**||**G-Ref**|          |
> |:---------:|:---:|:---:|:---:|:---:|:---:|:---:|:---:|:--------:|
> |           |**mIoU**|**AP**|**mIoU**|**AP**|**mIoU**|**AP**|**mIoU**|  **AP**  |
> |     1     |**55.8**|**77.1**|**42.2**|**49.3**|**43.1**|**51.0**|**51.6**| **59.3** |
> |     2     |47.2|72.7|37.0|44.2|36.7|45.2|44.2|   53.8   |
> |     5     |46.0|71.1|36.4|43.5|36.0|44.3|43.6|   52.6   |
> |    15     |44.5|69.7|35.6|43.0|35.2|43.8|42.7|   51.8   |

---

### Official Review · Reviewer_qf5P · 2025-10-31

**Soundness:** 2
**Presentation:** 3
**Contribution:** 2
**Rating:** 2
**Confidence:** 4

**Summary:**

Authors propose a framework for text-based image segmentation that adapts Latent Diffusion Models (LDMs) using Rectified Flow (RF). They learn direct, deterministic mappings from image latents to mask latents without using stochastic noise to calculate intermediate time-step latents. They also introduce a Segment Anything (SAM) based mask refinement process, dynamic mask selection during training, and an adaptive one-step sampling for single-step inference.
Experiments on segmentation datasets show improved performance. Ablations also verify their selected design choices.

**Strengths:**

1. Interesting annotation refinement strategy.
2. One-step sampling makes sense give the deterministic nature of segmentation.
3. Valid analysis on how noise addition conflicts the nature of segmentation tasks.

**Weaknesses:**

**1. Inadequate related work discussion for mapping image to mask**:
This has been explored in numerous prior work (see paper below). These works are not discussed at all, please discuss these in related work section.
  - SemFlow (NIPS '24): https://arxiv.org/pdf/2405.20282
  - Section 3.2 in Method seems identical to this earlier paper (i.e. even uses same pretrained weights). Please distinguish how the proposed method differs.

**2. Inadequate related work discussion for adaptive one-step sampling**: Several prior works explore these ideas, please discuss them.
  - https://arxiv.org/pdf/2509.00036v1
  - https://openreview.net/pdf?id=YJ1My9ttEN (ICLR 2025)

**3. SAM labels dependency:** The RDS component appears to depend heavily on ability of SAM to generate good masks.
  - This becomes a weakness of the method, especially for domains where SAM may under perform. Exploration of such domains maybe interesting.
  -    The authors discuss negatives of human annotated masks (as polygon). However, are GT masks in the evaluation datasets collected in the same way (human annotated polygon masks)?

**4. Overfit to dataset annotation style**
 - Is this arising due to the noise-free training strategy of the proposed framework?
  - Would re-introducing noise fix this issue, instead of using the SAM based setup?

**5. Compute Budget**
  - What is the additional compute costs for training proposed model?
  - Is their a change in inference costs? How does this compare to prior work?

**6. Baseline details missing**
  - Implementation details of the reported baselines are missing.
  - Are the baselines pretrained on the same data as SD1.5?
  - How does proposed method compare to similar prior work such as SemFlow?


* SemFlow (NIPS 24): https://arxiv.org/pdf/2405.20282

**Questions:**

See weaknesses.

Compare against the SemFlow method that is very similar to author's approach?

---

> ### Author Response · Authors · 2025-11-25
>
> Thank you for the constructive feedback. Due to the character limit, we have provided concise responses here. We welcome any follow-up questions if further clarification is needed.
>
> # Comparison with SemFlow
> We clarify the distinctions between our method and SemFlow from two key perspectives:
> 1. Task Nature & Theoretical Formulation
>   - SemFlow: It targets a bidirectional mapping between unconditional segmentation and conditional generation. It lacks cross-modal capabilities (no text interaction) and is limited to fixed categories. Theoretically, a conflict exists between the determinism required for segmentation and the diversity required for generation. SemFlow introduces "finite perturbation" (adding noise to masks) to aid generation, which fundamentally compromises the deterministic precision needed for segmentation.
>   - Ours: While also based on Rectified Flow, our method focuses exclusively on text-driven segmentation. We support arbitrary text/referring inputs via lightweight fine-tuning. Unlike SemFlow's compromise, we optimize for precision: (1) We use background anchoring to correct the velocity magnitude ($|v|$) for one-step sampling; (2) We design the RDS module to mitigate the limitations of using generative models for discriminative tasks, reducing dependency on annotation styles.
> 2. Empirical Performance
>   Our method achieves a **zero-shot** mIoU of 39.7 on COCO-Stuff, surpassing SemFlow's fully trained result of 38.6. Detailed comparisons have been updated in Table 3 of the revised PDF.
>
> # Discussion on A-FloPs and AFM
> Thank you for suggesting a discussion of prior work on adaptive one-step sampling. A-FloPs accelerates general sampling via trajectory reparameterization, while AFM focuses on preserving high-frequency details in physical domains.
> In contrast, our method is explicitly tailored for segmentation. We leverage the prior that mask backgrounds are strictly zero-valued. By aligning low-luminance regions to "pure black" in the latent space, we create an anchor to correct the velocity magnitude $|v|$ —a key challenge in one-step sampling.
> Thus, our task-specific optimization is orthogonal to the general acceleration of A-FloPs and addresses a disjoint problem from AFM. We will incorporate this discussion into our revision.
>
> # SAM labels dependency
> We would like to address your concerns regarding "the dependency on SAM labels" with the following 2 points.
> 1. Model-Agnostic Framework & Small Object Strategy
>   - Our framework is model-agnostic and not strictly coupled with SAM; it can be swapped for other correctors.
>   - We tackled SAM's ineffectiveness on small targets by implementing a "Crop-Upscale-Refine-Paste back" strategy, ensuring high-precision refinement for even the finest details.
> 2. Strict Fairness in Evaluation
>  - The mask correction is applied exclusively to the training set. The test set remains completely unaltered. This ensures strict adherence to standard benchmarks, demonstrating that our model learns more robust features from refined training data without "cheating" on the test set.
>
> # Overfitting to Annotation Styles
> The overfitting to annotation styles is independent of SAM; rather, it stems fundamentally from the distinct learning objectives of discriminative and generative models.
>   - Traditional segmentation models optimize a pixel-wise Cross-Entropy loss. At object boundaries, where uncertainty is high, the model tends to minimize loss by predicting smooth probabilities (e.g., ~0.5), effectively "averaging" the boundary. This results in smoother, albeit blurrier, transitions.
>   - Generative models (our approach) excel at modeling the data distribution and preserving high-frequency details. Consequently, when trained on coarse, polygon-based annotations, the model faithfully learns the specific geometric distribution of these polygons rather than a smoothed approximation. It "memorizes" the sharp, jagged artifacts inherent in the ground truth.
>
> Introducing noise likely exacerbates overfitting by entangling specific annotation biases (e.g., polygon styles) with the sampling process. This causes the model to associate random noise with these styles during inference, leading to oscillating edges and inconsistent predictions rather than stability.
>
> # Compute Budget
> We detail the inference speed comparison of our proposed model against prior work. Tested on an RTX 3090 (avg. 100 runs), the speed ranking is LD-ZNet $\gtrsim$ Ours $\gg$ SemFlow. We are only marginally slower than LD-ZNet due to the overhead of the standard SD VAE decoder (designed for RGB). Utilizing a lightweight, mask-specific VAE would eliminate this redundancy, allowing our method to surpass LD-ZNet.
>
> |**Method**|**Inference Time (s)**|
> |:---:|:---:|
> |LD-ZNet|**0.17**|
> |SemFlow|1.09|
> |**Ours**|0.19|
>
> # Baseline details
> Yes, all SD-based baseline models (e.g., Ld-ZNet) utilized in this paper were pre-trained on the same dataset as Stable Diffusion v1.5.

---

### Comment · Area_Chair_aE3t · 2025-11-25
**Discussion with Authors**

Dear Reviewers,

The authors have diligently provided responses to your questions and concerns. I request you to please review the authors' responses, acknowledge that you have read them and actively engage with them in further discussion as needed.

This discussion period, with the authors, will end on December 2, 2025 (AoE). However, I request that you not wait until the last minute and actively engage with the authors early.

Best, AC

---

### Meta-Review · Area_Chair_Fzam · 2026-01-08

**Summary:**

This submission proposes a Rectified Flow-based text-based segmentation method that learns bi-directional mappings between images and segmentation masks in the latent space. The paper received reviews from four reviewers, resulting in divergent scores, i.e., three negative scores and one positive score. The core concerns related to a lack of technical innovation and comments on incremental novelty (i.e., the connection with SemFlow), the fairness of the evaluation, and multiple clarity issues.

After the rebuttal, the authors added several comparisons/ablations and clarified fairness (i.e., SAM refinement only on training). However, the dominant negative reviews remain driven by conceptual novelty, so the overall recommendation still trends toward rejection.

**Reviewer Concerns:**

Addressed:
1. Missing relevant baselines (VPD/ADPP) and single/multi-step ablation: Authors added VPD/ADPP comparisons on RefCOCO/+/G-Ref and provided 1/2/5/15-step ablations, supporting the “one-step works best” claim.
2. Fairness of RDS / SAM usage: Authors clarified that SAM refinement is applied only to the training set and reported an ablation indicating the core method remains competitive even without RDS.
3. SemFlow distinction (partially): Authors articulated task-level differences (text-driven vs fixed-category/unconditional) and added a direct performance comparison; this reduces ambiguity, though does not fully eliminate similarity concerns.
4. Compute/inference cost: Provided inference-time table (LD-ZNet vs SemFlow vs Ours) and noted the SD VAE decoder overhead.

Still Outstanding:
1. Incremental contribution: Even with clarifications, multiple reviewers view the method as engineering integration of known components (RF + LoRA + SAM + inference heuristic), with limited conceptual advance beyond closely related prior work.
2. Dependency and robustness of SAM refinement: There is still limited evidence on failure modes/cross-domain cases where SAM (or any refiner) underperforms.
3. Mask fidelity limits of SD VAE: The authors acknowledge potential thin-structure loss but do not empirically validate the impact or mitigation.

**Reviewer Scores:**

- Reviewer qf5P: remains 2
- Reviewer qbmG: remains 2
- Reviewer hGeg: would have changed to a lower score.
- Reviewer MSAt: remains 2

---

### Decision · Program_Chairs · 2026-01-26

Reject